# Postencephalitic Parkinsonism: Unique Pathological and Clinical Features—Preliminary Data

**DOI:** 10.3390/cells13181511

**Published:** 2024-09-10

**Authors:** Sabrina Strobel, Jeswinder Sian-Hulsmann, Dennis Tappe, Kurt Jellinger, Peter Riederer, Camelia-Maria Monoranu

**Affiliations:** 1Institute of Pathology, Department of Neuropathology, University of Wuerzburg, 97080 Wuerzburg, Germany; sabrina.strobel@uni-wuerzburg.de; 2Department of Medical Physiology, University of Nairobi, P.O. Box 30197, Nairobi 00100, Kenya; drjsian@gmail.com; 3Bernhard Nocht Institute for Tropical Medicine, 20359 Hamburg, Germany; tappe@bnitm.de; 4Institute of Clinical Neurobiology, A-1150 Vienna, Austria; kurt.jellinger@univie.ac.at; 5Clinic and Policlinic for Psychiatry, Psychosomatics and Psychotherapy, University Hospital Wuerzburg, 97080 Wuerzburg, Germany; peter.riederer@uni-wuerzburg.de; 6Department of Psychiatry, University of Southern Denmark, 5000 Odense, Denmark

**Keywords:** postencephalitic parkinsonism, Parkinson’s disease, iron pathology, Lewy bodies, neurofibrillary tangles, cognitive disturbances, α-synuclein, tau protein

## Abstract

Postencephalitic parkinsonism (PEP) is suggested to show a virus-induced pathology, which is different from classical idiopathic Parkinson’s disease (PD) as there is no α-synuclein/Lewy body pathology. However, PEP shows a typical clinical representation of motor disturbances. In addition, compared to PD, there is no iron-induced pathology. The aim of this preliminary study was to compare PEP with PD regarding iron-induced pathology, using histochemistry methods on paraffin-embedded post-mortem brain tissue. In the PEP group, iron was not seen, except for one case with sparse perivascular depositions. Rather, PEP offers a pathology related to tau-protein/neurofibrillary tangles, with mild to moderate memory deficits only. It is assumed that this virus-induced pathology is due to immunological dysfunctions causing (neuro)inflammation-induced neuronal network disturbances as events that trigger clinical parkinsonism. The absence of iron deposits implies that PEP cannot be treated with iron chelators. The therapy with L-Dopa is also not an option, as L-Dopa only leads to an initial slight improvement in symptoms in isolated cases.

## 1. Introduction

Parkinson’s disease (PD) is a heterogenic disease demonstrating various clinical phenotypes, genetic predispositions, and pathobiochemical pathways [1]. There is agreement that a great variety of triggers exist which are suggested to generate PD [2]. Such triggers involve viruses, which may cause a subtype of parkinsonian disorder including postencephalitic parkinsonism (PEP). The 1918 influenza A virus was discussed [3,4,5] but never proven to be cause of PEP [6,7,8]. However, there are also other viruses that have been thought to be causal for a virus-induced type of transient and/or consistent parkinsonism, such as avian influenza viruses, flaviviruses encompassing West Nile, Japanese B encephalitis, or St. Louis encephalitis viruses, the hepatitis C virus, HIV, and, recently, SARS-CoV-2 [5,9,10]. So far, the etiologic pathogen for PEP is unknown [8].

PEP shares many features with the idiopathic form of PD. There are only a few clinical descriptions of symptoms in PEP, which indicate motor disturbances, dystonia, disability, late deterioration mainly involving motor function [11], parkinsonism, and resting tremor in the three PEP patients studied [12] and parkinsonism in the one PEP patient studied by Caparros-Lefebvre et al. [13]. Of note, sparing of intellect [11], mild visuoconstructional memory deficit in one patient [13], three well-oriented cases of PEP until a few weeks before death with very mild cognitive impairment in only one of these patients [12], and no dementia in PEP patients studied by Wong et al. [14] have been described. Wenning et al. [15] describe six patients with clinicopathologically confirmed PEP, aged at death between 65 and 77 years, and a duration of clinical parkinsonian symptoms of 22 to 32 years. All suffered from oculogyric crises, akinesia, generalized rigidity, postural instability, and speech disturbances; four showed rest tremor, four vertical supranuclear gaze palsy, and two eyelid apraxia. Four developed visual hallucinosis and paranoid psychoses; five showed moderate memory deficits, abnormal attention, and frontal lobe dysfunctions with decreased fluency and disinhibition.

In contrast to the clinical picture neuropathological changes (degeneration of neuromelanin-containing dopaminergic neurons of the substantia nigra (SN) in PEP are not resembling idiopathic PD [16,17] as, in PEP, the entire SN is undergoing a massive degeneration, while, in idiopathic PD, the loss of neurons in the ventrolateral part of the SN is not a total one. Immunohistochemical studies of the PEP brains revealed diffuse disseminations of tau-positive neurofibrillary tangles (NFTs) and less frequent neurofibrillary threads. The NFTs were usually 3R and 4R tau-positive, with no differences in the immunohistochemical profile from those in Alzheimer’s disease (AD) [18]; only in the hippocampus, NFTs showed more frequent 4R tau positivity (as, e.g., in progressive supranuclear palsy (PSP). Tau-positive astroglial or oligodendroglial inclusions in white matter were rarely observed, while tufted astrocytes (characteristic hallmarks of PSP) were not observed. Furthermore, tau-positive granules in white matter in Guamanian parkinsonism–dementia complex (PDC)/amyotrophic lateral sclerosis (ALS) and argyrophilic grains were not observed in the PEP cases.

### 1.1. Although Similar to Idiopathic PD, PEP Shows Significant Differences

(1) Clinical: Onset of symptoms in a younger age group, including children and adults aged 25–40 years [19]. Significantly slower progression of parkinsonian symptoms in PEP (lasting between 17 and 33 years [18]). Rare occurrence of the characteristic resting tremor of PD. Progression of the disease in discontinuous spurts. Oculogyric crises and ophthalmoplegia. A prior history of encephalitis lethargica [20].

The age of onset of the encephalitis lethargica (EL), the interval to the development of extrapyramidal symptoms in the PEP cases included in this study, and information about the response to L-Dopa therapy is summarized in Table 1 and was published in our previous studies [8,15].

(2) Morphological: Total absence of α-synuclein pathology in PEP that therefore does not belong to the group of synucleinopathies [18,21]. PEP shows widespread NFTs, predominantly involving subcortical and brainstem areas [22,23,24,25], as well as in the hippocampus and entorhinal cortex, but only few NFTs in the frontal and temporal cortex [18,26], the majority of cases corresponding to Braak NFT stage III or IV. Since Aß immunohistochemistry was negative in the majority of PEP cases, and cerebral amyloid angiopathy (CAA) was not observed, PEP can be classified as “pure” tauopathy [18], which differs considerably from AD.

One of the major pathological processes discussed for PD, iron-induced oxidative stress [27], has not been studied in detail in PEP. Therefore, we here provide information about the possible role of iron in PEP pathology.

### 1.2. Iron and Neurodegeneration

Iron executes a broad range of vital roles in physiological processes including hemoglobin-related oxygen transport, transfer of electrons in the mitochondrial respiratory transport chain, and deoxyribonucleic acid synthesis [28]. It is regulated by the interplay between the hepatic hormone, hepcidin, the iron transporter, transferrin, and its chief iron exporter, ferroportin [29]. Hepcidin manages the iron flow into the plasma and extracellular fluid by instigating proteolysis or endocytosis of ferroportin [30]. Furthermore, ferroportin can be regulated transcriptionally and post-transcriptionally [31].

The maintenance of iron homeostasis is key to circumventing conditions/diseases related to deficiency or surplus concentrations of iron. Iron deficiency is the most commonly related to anemia [32,33]. Conversely, an iron overload can exert a more sinister consequence, particularly since excessive iron storage over a long period in major organs such as the liver can lead to cirrhosis or even organ failure [34]. Additionally, cellular iron overload can elicit events such as oxidative stress via the production of reactive oxidative/nitrative species, mitochondrial dysfunction, activation of inflammatory mechanisms, and iron-responsive protein-regulated cytotoxic ferroptosis [35] and disturb cellular biochemical processes, for instance, hindering ionic transport, resulting in permanent cellular damage and in cell destruction [36]. Therefore, the notion is of interest that iron in the SN is stored mainly by ferritin in the glia, while the dopaminergic neurons of the SN neuromelanin show a high iron storage capacity [37,38].

Repeated findings in the literature have reported iron depositions in cerebral regions with pathologic changes, suggesting that it may be closely linked to the pathogenesis of many degenerative disorders. Indeed, an elevation in total iron in the SN in some basal ganglia disorders including PD [27,39,40], multiple system atrophy, and progressive supranuclear palsy [40]. In contrast, the total iron content was elevated only in the striatum in Huntington’s disease [40]. Other neurological diseases that display abnormal iron augmentation in the caudate–putamen and globus pallidus include AD [41], multiple sclerosis (Burgetova et al. 2020; [42]; Schmalbrock et al. 2016), and ALS [43]. Additionally, a collection of inherited diseases (neurodegeneration with brain iron accumulation, NBIA) exhibiting neurodegeneration coupled with different extents of atypical iron deposition in the brain all share basal ganglia motor symptoms and cognitive decline [44]. Interestingly, inherited movement disorder neuroferritinopathy shows similar diffuse iron accumulation to that reported due to advancing age [45].

An elevation of cerebral iron also occurs as an age-related phenomenon, and in vivo magnetic resonance imaging (MRI) has demonstrated regional heterogeneity of the iron depositions [46,47]. This notion is illustrated by the predilection of elevated iron levels with advancing age in the basal ganglia regions including the SN, caudate nucleus, putamen, and globus pallidus and also some areas in the cortex [48,49]. It therefore appears that the tendency for the neurons to accumulate iron is dependent on the anatomical site and is disease-specific. It is an element with diverse influences on neurodegenerative disorders [50]. Although often conflicting results of brain iron content have been reported, this may be related to the method of assessment, such as direct methods performed on post-mortem tissue or indirect methods, for example, in vivo MRI [51].

## 2. Materials and Methods

### Sampling and Processing

We used formalin-fixed and paraffin-embedded tissue from midbrain sections (SN, pars compacta) from neuropathologically characterized PD with different Braak Lewy Bodies (LBs) stages and age-matched controls collected from the Brain Bank Center Wuerzburg for analysis. These brains were obtained with the consent of the next of kin and according to the guidelines of the national and local ethics committees. This study was approved by the Local Ethics Committee of the University of Wuerzburg (internal application number 99/11) and was performed in keeping with the ethical standards described in the most recent version of the Declaration of Helsinki. Cases with PEP, collected from the Institute of Clinical Neurobiology, Vienna, were analyzed in parallel for comparison. Neuropathology of the five cases with PEP (65 to 77 years old, all Caucasians) was featured by severe diffuse neuronal depletion and gliosis in the SN, locus coeruleus, subthalamic nucleus, midbrain raphe, pontine tegmentum, periaqueductal gray matter, and many brainstem nuclei including the oculomotor nucleus complex, cranial nuclei of glossopharyngeal and vagus nerve, pontine tegmentum, and reticular formation, with relative preservation of the red nucleus. Histology in one case (66 years) showed a Braak NFT stage IV with amyloid plaques in the hippocampus; all others were at a Braak NFT stage III, with only a few senile plaques in one single case. This neuropathological pattern explains the rather mild cognitive disturbances of these patients, predominantly of their short memory. Only in the one case, aged 66 years, the deficit in short memory was notable. In all other cases, psychotic disturbances with visual hallucinations and paranoid delusions, respectively, frontal lobe symptoms, were dominant.

Three groups were created: controls (n = 11; Braak LB stage 0, no other neuropsychiatric disease), PD (n = 18; Braak LB stages II–VI), and PEP (n = 5). The ages at death ranged from 49 to 91 years in the control group, 58 to 96 years in the PD group, and 65 to 77 years in the PEP group. The male–female ratios were 5:6 among controls, 11:7 in the PD group, and 2:3 1:4 in the PEP group (Table 1).

In order to assess iron deposition in the tissue, Perls Prussian blue reaction (non-heme iron, Fe^3+^) was used. The deparaffinized tissue section is immersed in a solution of potassium ferricyanide (1–2%) and hydrochloric acid (1–2%) in a ratio of 1:2. The iron in the section is ionized by the hydrochloric acid. It now has a high affinity for ferrocyanide and displaces the potassium. Ferri-ferro-cyanide is formed, a salt that is difficult to dissolve. The preparation is then rinsed with distilled water and counterstained with Kernechtrot, which contains aluminum sulfate as a mordant. In the preparation, the nuclei appear red and the iron deposits (e.g., hemosiderin) blue.

## 3. Results

We were able to determine differences in iron deposition between the groups. In the control group, iron was predominantly found in perivascular spaces and deposited in cells other than neurons (e.g., oligodendrocytes and microglia), whereas, in the PD group, all cases except two showed iron depositions in nigral neurons containing neuromelanin (Figure 1).

In the PEP group, the SN was almost completely devoid of neurons containing neuromelanin, and iron depositions were not seen, except for one case with very sparse perivascular deposits (Figure 1) (Table 2).

The appearance of iron deposits in post-mortem brain tissue from PD suggests that nigral iron accumulation may choreograph a fundamental role in the destructive pathways (via production of hydroxyl radicals and oxidative stress) of neuromelanin-containing neurons and also in the motor deficits and motor abnormalities [52]. Indeed, it has been suggested that the increase in iron content may correlate with a degree of SN dopaminergic neuronal destruction, which, in turn, impacts the motor deficits/abnormalities in PD [53]. Extrapolation of this concept would imply that parkinsonian-specific neuronal iron localization contributes to the manifestation of selective motor features. However, this notion is challenged by disorders such as PEP. PEP has been reported after viral infections with influenza A virus, West Nile, St. Louis, Varicella zoster, and Japanese B encephalitis viruses [5,54]. It therefore appears that the viral infection subsequently has the propensity to result in the manifestation of parkinsonism. Paradoxically, although PD and PEP share many motor deficits, such as, e.g., bradykinesia, resting tremor, and rigidity, they are quite diverse neuropathologically. The varied neuropathological features in the two disorders range over abnormal accumulation of different proteins, selective loss of neurons, deposition of iron, and other features [8,10].

### Neuropathological Diversity in PD and PEP

PD is predominantly an α-synucleinopathy; however, ß-amyloid pathology (plaques) and tau aggregates (neurofibrillary tangles) have also been found in about 50% of PD brains in the SN [17,55], thereby implicating a credible association between PD and tauopathology. Perhaps, the hyperphosphorylated tau proteins may co-aggregate with the misfolded α-synuclein, especially in neuromelanin-containing neurons susceptible to LB formation. As such, the neuropathological findings at post-mortem exhibit a more complex story, and a pure pathology is not such a common occurrence [56]. Indeed, more frequently, a double- or triple-pathology is found [57], thereby reflecting diversity in the destructive mechanisms with some common pathways. Fascinatingly, α-synuclein can adopt various structural conformations depending on the environment [2,58]. Furthermore, since these different α-synuclein structures have distinct functions, this may bestow the ability for it to colocalize with other proteins such as tau and/or ß-amyloid in neurodegenerative diseases [59]. Indeed, this concept is supported by the presence of Lewy-related pathology (mainly consisting of α-synuclein) in AD [60]. More importantly, Twohig and colleagues (2018) reported an association between α-synuclein concentration in the cerebrospinal fluid, *APOEε*4 risk allele, and amyloid beta deposits in early AD [61].

In contrast, PEP exhibits almost exclusively tau pathology only and does not show α-synuclein pathology and LB pathology [18,21]. Under physiological conditions, tau proteins stabilize microtubules. However, if these proteins are hyperphosphorylated, they break away from the microtubules to form oligomers which subsequently develop into NFTs that disrupt neuronal function [62]. NFTs in PEP have been observed in the remaining nigral neurons and also the brain stem in progressive supranuclear palsy [63]. The similarity of the NFTs in PEP and AD [63] may be suggestive of common degenerative reactions. Free-radical-induced oxidative stress has been implicated in the formation of NFTs [64]. This notion has been supported by the presence of the oxidative/nitrative stress marker, nitrotyrosine, in the NFTs in AD brains [65]. Oxidative stress probably represents a common denominator for inducing dysfunction of protein homeostasis.

Tau proteins are also present in the pathological hallmark of PD, and LBs have been found in the SN and other selective brain areas in the disease [66]. These structures predominantly contain aggregates of misfolded α-synuclein proteins which disrupt axonal transport, leading to neuronal destruction [67]. The proximity of LBs to areas exhibiting cell loss supports the notion that they represent neuronal destruction and may be involved in the process. It has been suggested that the hyperphosphorylated tau proteins may contribute to the formation of LBs by associating with α-synuclein to augment its aggregation [62]. However, since only about half of the PD brains exhibit tauopathology, different pathogenesis and mechanisms for LBs may operate in other cases of PD [55]. Thus, advocating the multifactorial hypothesis as the cause of PD [68] supports the involvement of aging, environmental factors, coupled with some genetic predisposition. Additionally, other components include post-translational α-syn modifications, variation in LB precursors, and the selective vulnerability of specific brain regions (such as the SN). Conversely, LBs may serve a protective role, through their ability to sequester iron in the PD brain, as opposed to a destructive one [69,70]. This concept is supported by the presence of reactive iron in LBs present in the SN in PD [69]. The iron (III) form has a predilection for the production of cytotoxic hydroxyl radicals and thus oxidative stress, and, more importantly, it binds to α-syn, perhaps causing a molecular change that prompts its aggregation [71,72].

Alternatively, the brain iron overload may just be one of the many degenerative contributors, and it may just play a small part in the whole pathological cascade, and the rogue α-synuclein protein may be occupying the center stage, skillfully executing cellular catastrophe (Sian et al., 2015; Sian-Hulsmann and Riederer 2021). For instance, the presence of nigral LBs in asymptomatic PD or incidental LB disease without any changes in the iron content [73] suggests that perhaps the misfolded α-synuclein (LB) may have invoked oxidative stress reactions, which contribute to the depletion of the antioxidant glutathione (reduced form, GSH) in the SN of these subjects [73]. Its ability to function as ferrireductase (involved in the conversion of Fe (III) to Fe (II)) may contribute to dysregulation of the iron homeostasis and consequential elevation of nigral iron levels in PD [39,74,75] and neuronal destruction in iron-sensitive brain areas. The pathophysiological pathways in Parkinson’s disease and postencephalitic parkinsonism are shown in Figure 2.

## 4. Discussion

For the present study, histopathological staining of brainstem slides was used to identify the localization of iron deposits inside selective cell types. A distinctive pattern of cellular iron accumulation was observed in the three groups, whereby the control subjects represented non-pathological, probably age-related changes in cerebral iron. In the control subjects, the iron appears to be located mainly in the non-neuronal cells (microglia/oligodendrocytes) and perivascular spaces. In contrast, the iron deposits were present in the remaining nigral neurons and non-neuronal cells and to a lesser degree in the perivascular spaces for the PD group. The significant iron levels in the dopaminergic neurons are probably related to the presence of neuromelanin, which has a high iron-loading capacity [38,70]. There are some putative modes of iron sequestration by neuromelanin [76]. The neuromelanin may enter the autophagy process and bind with iron released through proteolysis from ferritin. On the other hand, neuromelanin may be associated with iron via iron-binding chaperone proteins.

Our findings are similar to those reported by Friedrich and colleagues (2021) [77]. Perhaps this may be suggestive of cellular redistribution of some of the iron from the non-neuronal cells/perivascular spaces to the neurons in the pathological state. The predominance of iron in the remaining surviving nigral neurons in PD demonstrates its importance in the pathomechanisms operative in the disease process, probably as a secondary phenomenon. Furthermore, this is unlikely to be an age-related change, as exhibited by the absence of iron in the neurons in the control subjects.

The presence of iron deposits in non-neuronal cells, particularly the microglia in both the control subjects and PD, may be of key importance in the neurodegenerative processes. In the diseased state, the progressive neuronal loss may serve as a stimulus to activate microglia. Additionally, microgliosis may induce an imbalance leading to a greater secretion of pro-inflammatory (such as tumor necrosis factor-α, interleukin-1β) than anti-inflammatory factors, resulting in chronic neuroinflammation [78]. Furthermore, these pro-inflammatory cytokines have the propensity to disturb iron homeostasis via endocytosis of ferroportin from macrophages, resulting in iron deposition in the brain [30,79]. Abnormal iron accumulation has been observed in the 6-hydroxydopamine (6-OHDA;) hemi-parkinsonian rat model, for which three possible causes were discussed: a compromised blood–brain barrier (BBB), abnormal expression of ferritin, and neuroinflammation. Alterations of the BBB were identified by gadolinium-enhanced MRI, and detection of extravasated IgG suggested transport through a leaky BBB. Presence of iron following degeneration of dopaminergic cells by MRI revealed hypointense signals in the substantia nigra. Histologically confirmed iron deposition was closely associated with microglia and increased levels of L-ferritin, indicating a possible role for both changes in brain accumulation and dopamine neurodegeneration [80]. The characteristic elevation of cerebral iron in PD may also be related to the significant depletion (almost 50%) of ferritin in oligodendrocytes [77]. There is compelling evidence indicative of iron dyshomeostasis in the pathogenesis of PD, and this includes the dysfunction of iron-metabolism-related proteins, resulting in abnormalities in its transport and storage [52].

Thus, these changes/presence in the pathological state may serve as a “double hit” whereby the iron accumulation and the activated microglial cells [81,82] may result in augmenting free-radical-mediated oxidative stress and consequential dopaminergic neuronal destruction [5]. Additionally, these two processes may even exacerbate α-synuclein aggregation and LB formation [83].

The physiological function of perivascular spaces appears to be related to the maintenance of brain health. However, studies using magnetic resonance imaging of perivascular spaces have shown that they increase with age and adopt a more pathological role, which may imbalance cerebral vascular hemodynamics and favor a neurodegenerative pathology [84]. Therefore, it seems that the perivascular spaces represent pathways for the cerebral vascular transport of iron and that the elevation of iron in these spaces may be a result of age-related dilation and dysfunction. The presence of iron deposits in the perivascular spaces is suggestive of some degree of perivascular and micro-vessel malfunction, and, thus, extrapolation of this would suggest the predilection of the SN to age-associated small-vessel disease and neurodegeneration [85,86]. However, a study of iron perivascular accumulation in other brain regions is warranted to make a fair comparison and to determine areas that may demonstrate a similar predisposition or trend. More importantly, given that perivascular spaces serve a crucial role in priming neuroinflammation in the brain, they may contribute/be associated with neuronal cellular damage [87]. Interestingly, studies using magnetic image resonance have shown an elevation in perivascular volumes coupled with blood–brain barrier destruction and the appearance of contrast-enhancing lesions [87]. Perhaps, the perivascular spaces prompted the neuroinflammatory cascade, which disrupted the blood–brain barrier, and manifestation of the contrast-enhancing lesions. A recent study suggested that, since perivascular spaces burden correlates with PD motor deficit/abnormalities, they may be employed as biomarkers in the early stages of the illness [88].

Initially, the BBB has been thought to remain intact during neurodegenerative diseases, but accumulating evidence has suggested the possible association of BBB dysfunction with PD pathology and especially in virus-induced parkinsonism disorders [5]. Among the myriad of pathogenic mechanisms leading to neurodegeneration in PD, tight junction alterations and neurovascular unit dysfunctions, which ultimately cause altered BBB permeability, have been documented [89]. This can result in dysregulated ionic homeostasis, impaired transport of nutrients, and accumulation of neurotoxins that ultimately lead to irreversible neuronal loss [90]. Earlier in vivo studies demonstrated BBB dysfunction in the parkinsonian midbrain [91] and in the putamen [92]; this nigrostriatal BBB opening in PD was confirmed recently by 18F-Choline-PET [93]. According to a systemic meta-analysis, biofluid markers suggest BBB disruption and neurodegenerative co-pathology involvement in Lewy-body-spectrum diseases [94], which can be explained by the impact of α-synuclein aggregates on the members of the neurovascular unit (perivascular astrocytes, microglia, and endothelial cells) [95]. BBB damage is also observed in tauopathies that lack amyloid-β overproduction, suggesting a role for tau in BBB damage. This is driven by chronic neuroinflammation promoting structural changes in capillaries such as fragmentation, thickening, atrophy of pericytes, accumulation of laminin in the basement membrane, and increased permeability of blood vessels to plasma proteins, confirming the role of tau protein in BBB structural and functional changes [96].

The absence of iron deposits in PEP disqualifies the theory that suggests that clinical parkinsonian features are a consequence of raised iron content in the SN neurons [97]. It appears highly likely that it is the loss of the dopaminergic neurons in the SN that produces the motor abnormalities. Indeed, this idea is supported by the reversal of most motor deficits in PD after the introduction of the dopamine precursor 3,4-dihydroxy-l-phenylalanine (L-DOPA)/carbidopa resp. L-DOPA/benserazide treatment [98,99,100]. The benefit of L-dopa/decarboxylase inhibitor treatment can be attributed to its ability to restore the disease-depleted striatal dopamine levels. Similarly, favorable responses to L-dopa/carbidopa treatment in PEP patients were reported, although the doses required careful titration to avoid unwanted side effects (such as dyskinesias) [101], thereby endorsing the importance of the nigral dopamine deficit in the manifestation of parkinsonian-like motor symptoms.

Strangely, in PEP there is also a complete disappearance of iron from the non-neuronal cells and perivascular spaces in the SN. This is probably related to the pathogenesis of the illness and further endorses a different mechanism(s) for dopaminergic neuronal destruction compared to PD. Furthermore, this concept is supported by the difference in pathological markers between the two disorders, α-synuclein pathology/LBs in PD and tauopathology/NFTs in PEP [38].

The occurrence of gliosis and, in some cases, residual infiltration of mononuclear cells coupled with neuronophagia support the occurrence of some degree of neuroinflammation in the disease. Although this is probably an early and primary event, evidence for inflammation at autopsy is limited or absent. Also, the infectious agent (virus causing encephalitis lethargica) is absent at autopsy [8]. Since the related infection occurs significantly before the manifestation of clinical parkinsonism, this suggests that perhaps neuroinflammation does not play an active role in the pathogenesis of the disorder or that, alternatively, it initiates the neurodegenerative cascade via cytotoxic processes such as oxidative stress. The latter appears to be a viable possibility since these stressor processes are ascribed to the nigral neuronal destruction observed in the disease. Unfortunately, there are limited data regarding antioxidants such as glutathione or even activities of antioxidant enzymes in the SN in PEP, thereby making it difficult to deduce the degree of oxidative stress in the pathogenesis of the illness.

Perhaps some cellular protective treatment against oxidative stress can be advocated in the early stages of the disease (PEP and PD), such as antioxidant agents to combat the protein dysregulation/neuronal cell destruction/reduction of cytoplasmic dopamine dysfunction and fluctuations in intracellular calcium levels, since they are involved in dopamine release [102,103].

Viral-infection-induced parkinsonism may induce PD features either immediately (para-infectious) or subsequently (post-infectious) [104], although the two mechanisms are not mutually exclusive and can co-exist. The long period (over a decade) between the contraction of the viral agent and the manifestation of the parkinsonian features in PEP supports the post-infectious pathway and appears to dilute the direct involvement of the infectious agent, since it appears highly unlikely that the infectious agent remains dormant or marginally active for many years eliciting selective neuronal destruction. Nevertheless, the delayed post-infectious parkinsonian mechanisms may elicit neuronal destruction through autoimmune processes induced by several potential factors, including epitope spreading, bystander activation, polyclonal B-cell activation, or molecular mimicry [9]. Thus, the occurrence of PEP extrapyramidal symptoms post-viral infection supports some integral rather than remote involvement.

However, in PD, it appears that a delayed period between the onset of nigral neuron destruction and the manifestation of motor symptoms may be related to another phenomenon such as brain plasticity. Brain plasticity at both cellular and synaptic levels probably contributes to the long preclinical period before the manifestation of symptoms in PD (longer than 6 years, [105]). Perhaps, the loss of the striatal dopamine leads to many adaptive/compensatory cellular changes as a coping mechanism; however, when these are overwhelmed, the motor abnormalities and other symptoms ensue [106]. Indeed, it is well documented that neurons are versatile and can adapt to the dopamine deficit (to a certain degree and for a certain period of time) to preserve nigrostriatal tract and basal ganglia function via homeostatic plasticity processes [107].

Interestingly, both tau pathology in PEP [108] and α-synuclein pathology in PD [109] spread and propagate in a prion-like manner. Although the precise mechanism underlying the spread is unclear, it appears to occur via synaptic-associated neuronal tracts [109]. The anti-hyperglycemic drug, metformin, has been found to diminish the propagation of tau pathology in PS 19 mice by reducing tau hyperphosphorylation and improving learning and memory deficits [110]. Similarly, metformin has been shown to exert neuroprotective actions in 1-methyl-4-phenyl-1,2,3,6-tetrahydropyridine-treated mice (a model for PD) by blocking α-synuclein phosphorylation and increasing neurotrophic factors [111]. Thus, it has the potential to serve as a useful therapeutic drug in the early stages of tau and α-synuclein pathologies to reduce the progression of neurodegeneration and warrants further investigation.

## Figures and Tables

**Figure 1 cells-13-01511-f001:**
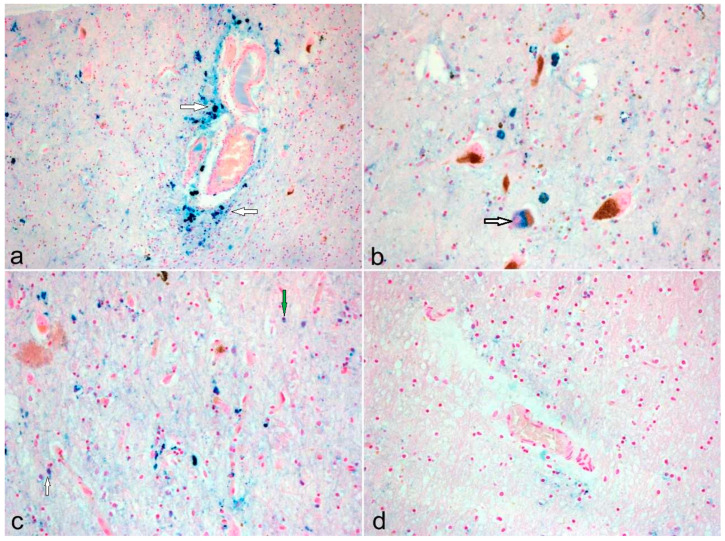
Iron depositions: (**a**) perivascular (arrows); (**b**) intraneuronal (arrow); (**c**) microglia (white arrow), oligodendrocyte (green arrow); (**d**) sparsely perivascular deposition; (**a**–**c**) PD cases; d PEP case (magnification 100×).

**Figure 2 cells-13-01511-f002:**
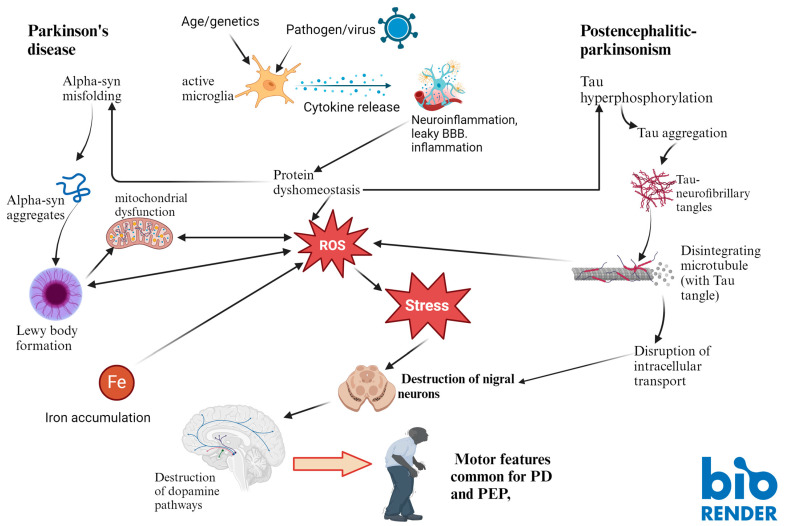
Pathophysiological pathways in Parkinson’s disease and postencephalitic parkinsonism.

**Table 1 cells-13-01511-t001:** Clinical data of the PEP cases included.

Case Number	Extrapyramidal Features	L-Dopa Response	Tendency to Fall
PEP 1	Bradykinesia, retropulsion, oculogyric crises	+	+
PEP 2	Tremor, hypomimia, gait disturbance	+	+
PEP 3	Gait disturbance	0	+
PEP 4	Sialorhea, rigidity, gait disturbance	+	+
PEP 5	Bradikinesia, tremor	0	+

PEP = Postencephalitic parkinsonism.

**Table 2 cells-13-01511-t002:** The results together with the demographic data of cases analyzed.

Case Number	Age at D/E/S	Gender	Braak (LB)	Localization of Iron	α-Syn	Tau
CO 1	80	m	0	p	−	−
CO 2	74	m	0	n, i, p	−	−
CO 3	65	m	0	i, p	−	−
CO 4	89	f	0	p	−	−
CO 5	91	m	0	i	−	−
CO 6	73	f	0	i, p	−	−
CO 7	49	m	0	i, p	−	−
CO 8	60	f	0	i, p	−	−
CO 9	67	f	0	i	−	−
CO 10	71	f	0	i	−	−
CO 11	62	f	0	i	−	−
PD 1	69	f	IV	n, i	+	−
PD 2	58	m	IV	n, i, p	+	−
PD 3	85	m	IV	i	+	−
PD 4	87	f	III	i	+	−
PD 5	70	f	IV	n, i, p	+	−
PD 6	72	m	V	i, p	+	−
PD 7	83	m	VI	n, i, p	+	−
PD 8	69	f	V	n, i, p	+	−
PD 9	72	f	IV	n, i, p	+	−
PD 10	96	f	IV	n, i, p	+	−
PD 11	58	m	VI	n, i, p	+	−
PD 12	80	m	IV	n, i	+	−
PD 13	63	m	II	n, i	+	−
PD 14	77	m	III	n, i, p	+	−
PD 15	81	m	IV	n, i	+	−
PD 16	87	m	VI	n, i, p	+	−
PD 17	58	w	III	n, i	+	−
PD 18	72	m	V	n, p	+	−
PEP 1	66/26/38	f	0	Ø	−	+
PEP 2	77/16/60	f	0	Ø	−	+
PEP 3	66/16/44	m f	0	Ø	−	+
PEP 4	65/11/32	f	0	p	−	+
PEP 5	69/17/45	m	0	Ø	−	+

CO = controls, PD = Parkinson’s disease, PEP = postencephalitic parkinsonism, D = death, E= encephalitis lethargica onset, S = onset of extrapyramidal symptoms, Ø = negative, i = iron intracytoplasmic in cells other than neurons (oligodendrocytes and microglia), p = iron perivascular, n = iron in neurons, α-syn = α-synuclein, Tau = tau depositions (neurofibrillary tangles).

## Data Availability

The original contributions presented in the study are included in the article, further inquiries can be directed to the corresponding author.

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
