# Peer review of "Postencephalitic Parkinsonism: Unique Pathological and Clinical Features—Preliminary Data"

_cells, 2024, doi:10.3390/cells13181511_

Round 1
Reviewer 1 Report
Comments and Suggestions for Authors
The manuscript (cells-3122179) reviewed the clinical findings and pathology of postencephalitic parkinsonism (PEP) as compared to classical Parkinson’s disease (PD). In particular, the article focused on iron pathology, including their qualitative observations in a cohort of autopsied brains from patients of PEP (n=5), PD (n=18) and matched healthy controls (n=11). Nigral neuronal iron depositions were found only in PD but not in PEP. The distinctive iron deposit differences in PEP vs PD were further discussed in combination with tau and α-synuclein pathology, respectively, in the two parkinsonian conditions.
Overall, the topic is interesting, in particular in the post-COVID-19 era when PEP might be of clinical interest. It is thus useful to provide readers an updated summary of the distinction between PEP and PD. However, more clinical and pathological information can be provided for the PD and PEP patients in Table 1, e.g., levodopa treatment, postmortem time, and tau and/or α-synuclein findings.
Author Response
Response to Reviewer 1 Comments
We thank you for your expert comments and the opportunity to respond to the criticisms of our manuscript, modifications are marked in the text as follows: parts highlighted in gray to be removed; paragraphs highlighted in yellow are newly added.
Comments and Suggestions for Authors
The manuscript (cells-3122179) reviewed the clinical findings and pathology of postencephalitic parkinsonism (PEP) as compared to classical Parkinson’s disease (PD). In particular, the article focused on iron pathology, including their qualitative observations in a cohort of autopsied brains from patients of PEP (n=5), PD (n=18) and matched healthy controls (n=11). Nigral neuronal iron depositions were found only in PD but not in PEP. The distinctive iron deposit differences in PEP vs PD were further discussed in combination with tau and α-synuclein pathology, respectively, in the two parkinsonian conditions.
- Overall, the topic is interesting, in particular in the post-COVID-19 era when PEP might be of clinical interest. It is thus useful to provide readers an updated summary of the distinction between PEP and PD. However, more clinical and pathological information can be provided for the PD and PEP patients in Table 1, e.g., levodopa treatment, postmortem time, and tau and/or α-synuclein findings.
Answer: We added a second Table including informations about the main symptoms and L-Dopa treatment/response for the PEP cases. For these cases postmortem time is not available. The tau and α-synuclein findings of the PEP cases were supplemented now. Former Table 1 is now Table 2.
As required by the editor, we reduced the number of self-citations in the references list: 3, 19, 22, 43, 44, 50, 60, 63, 79, 80, 82, 87, 90 were deleted.

Reviewer 2 Report
Comments and Suggestions for Authors
The major limitation of the study on Postencephalitic Parkinsonism (PEP) presented is the small sample size. The study included only 5 cases of PEP, which may limit the generalizability of the findings and the ability to draw robust conclusions about the pathology and clinical features of PEP. Larger sample sizes are typically preferred in research studies to increase statistical power and reliability of the results. The author need to explain it more.
Abstract is too short. Improve its background, question, objectives, methods, and implications.
Discussion part add 1 or 2 more figures.
Minor limitations of the study on Postencephalitic Parkinsonism (PEP) presented in the PDF file may include:
Lack of longitudinal data: The study may not have included longitudinal data on the progression of PEP symptoms over time, which could provide valuable insights into the disease course.
Limited diversity in patient demographics: The study may have lacked diversity in terms of patient demographics, such as age range, gender distribution, or ethnic backgrounds, which could impact the generalizability of the findings.
Potential selection bias: There may have been a selection bias in the recruitment of PEP cases, as the cases were collected from specific institutions, which could affect the representativeness of the sample.
Absence of functional assessments: The study may not have included detailed assessments of functional abilities or quality of life measures in PEP patients, which could provide a more comprehensive understanding of the impact of the disease on daily living.
Limited information on treatment outcomes: The study may not have provided detailed information on treatment outcomes or responses to therapies in PEP patients, which could be important for guiding clinical management strategies.
Addressing these minor limitations in future studies could enhance the depth and breadth of knowledge on Postencephalitic Parkinsonism.
Author Response
Response to Reviewer 2 Comments
We thank you for your expert comments and the opportunity to respond to the criticisms of our manuscript, modifications are marked in the text as follows: parts highlighted in gray to be removed; paragraphs highlighted in yellow are newly added.
Comments and Suggestions for Authors
- The major limitation of the study on Postencephalitic Parkinsonism (PEP) presented is the small sample size. The study included only 5 cases of PEP, which may limit the generalizability of the findings and the ability to draw robust conclusions about the pathology and clinical features of PEP. Larger sample sizes are typically preferred in research studies to increase statistical power and reliability of the results. The author need to explain it more.
Answer:
The expansion of the PEP group was also our greatest wish. Therefore, Prof. Tappe traveled all over Germany and Austria to collect cases, unfortunately without success. These are very old cases, which have been removed from the archives over time. We decided therefore to add: “preliminary data” to the manuscript title.
Abstract is too short. Improve its background, question, objectives, methods, and implications.
Answer: Thank you for this observation, we extended the abstract as follows (yellow). Gray to be removed.
Postencephalitic parkinsonism (PEP) is suggested to show a virus-induced pathology, which is different from classical idiopathic Parkinson’s disease (PD) as there is no α-synuclein/Lewy body pathology. However, PEP shows a typical clinical representation of motor disturbances. In addition, and compared to PD, there is no iron-induced pathology. The aim of this preliminary study was to compare PEP with PD regarding iron-induced pathology, using histochemistry methods on paraffin-embedded post-mortem brain tissue. In the PEP group iron were not seen, except for one case with sparse perivascular depositions. Rather, PEP offers a pathology related to tau-protein/neurofibrillary tangles, with mild to moderate memory deficits only. It is assumed that this virus-induced pathology is due to immunological dysfunctions causing (neuro)inflammation-induced neuronal network disturbances as events that trigger clinical parkinsonism. The absence of iron deposits implies that PEP cannot be treated with iron chelators. The therapy with L-Dopa is not the therapy of choice, as L-Dopa only leads to an initial slight improvement in symptoms in isolated cases.
Discussion part add 1 or 2 more figures.
Answer: We briefly considered adding pictures of the tau immunohistochemistry in PEP, but such figures, from the same cases have already been published, please see our previous publication which has been quoted in this regard (Reference 8 and 15).
In the above mentioned publication the first 5 cases from the table overlap with our cases. From case 6 no more tissue was available. For more informations regarding the PEP cases included, we added one Table more (Table 1 is now Table 2).
Minor limitations of the study on Postencephalitic Parkinsonism (PEP) presented in the PDF file may include:
- Lack of longitudinal data: The study may not have included longitudinal data on the progression of PEP symptoms over time, which could provide valuable insights into the disease course.
Answer: We obtained some clinical data from PEP cases data from the Austrian archives regarding major symptoms as well as L-Dopa and present those in the new Table 1.
- Limited diversity in patient demographics: The study may have lacked diversity in terms of patient demographics, such as age range, gender distribution, or ethnic backgrounds, which could impact the generalizability of the findings.
Answer: The age range and gender distribution was already included in the first version in Chapter 2: Material and Methods: age at death was 65 to 77 years, the male:female ratio 2:3. We verified the gender data and we detected an error: correctly is male:female ratio 1:4. All patients were Caucasians (were added in yellow).
Potential selection bias: There may have been a selection bias in the recruitment of PEP cases, as the cases were collected from specific institutions, which could affect the representativeness of the sample.
Answer: In our opinion, the fact that all PEP cases come from one and the same institution also has advantages, especially with regard to the uniform processing and storage of the tissue, a very important criterion that allows the cases to be compared directly. In fact, as already mentioned above, we could not find more PEP cases in other German or Austrian institutions and were very grateful to Prof. Jellinger for the 5 cases provided.
Absence of functional assessments: The study may not have included detailed assessments of functional abilities or quality of life measures in PEP patients, which could provide a more comprehensive understanding of the impact of the disease on daily living.
Answer: The onset of parkinsonism symptoms was in the fifties-sixties of the last century, informations about quality of life measures were not collected that time. What is known is that patients developed akinesia and were finally wheelchair-bound and, in part, also desorientated. Hallucinations and depression with paranoid delusions were also noted. It sounds like a poor quality of life.
- Limited information on treatment outcomes: The study may not have provided detailed information on treatment outcomes or responses to therapies in PEP patients, which could be important for guiding clinical management strategies.
Answer: Longitudinal clinical data from the PEP cases included in this study are summarized in Table 1
- Addressing these minor limitations in future studies could enhance the depth and breadth of knowledge on Postencephalitic Parkinsonism.
Answer:
In order to counteract these limitations, we would need significantly more PEP cases. We could also contact colleagues in neuropathology departments in other countries besides Germany and Austria for a future study.
As required by the editor, we reduced the number of self-citations in the references list: 3, 19, 22, 43, 44, 50, 60, 63, 79, 80, 82, 87, 90 were deleted.

Round 2
Reviewer 2 Report
Comments and Suggestions for Authors
No further issue.